# Quantification of toxic metals using machine learning techniques and spark emission spectroscopy

Seyyed Ali Davari[1,*] and Anthony S. Wexler[1,2]

[1]Air Quality Research Center (AQRC), University of California, Davis, 95616, Davis, USA
[2]Department of Mechanical and Aerospace Engineering, Civil and Environmental Engineering, and Land, Air and Water Resources, University of California, Davis, USA

### Abstract

The United States Environmental Protection Agency (US EPA) list of Hazardous Air Pollutants (HAPs) includes toxic metal suspected or associated with development of cancer. Traditional techniques for detecting and quantifying toxic metals in the atmosphere are either not real time, hindering identification of sources, or limited by instrument costs. Spark emission spectroscopy is a promising and cost effective technique that can be used for analyzing toxic metals in real time. Here, we have developed a cost-effective spark emission spectroscopy system to quantify the concentration of toxic metals targeted by US EPA. Specifically, Cr, Cu, Ni, and Pb solutions were diluted and deposited on the ground electrode of the spark emission system. Least Absolute Shrinkage and Selection Operator (LASSO) was optimized and employed to detect useful features from the spark-generated plasma emissions. The optimized model was able to detect atomic emission lines along with other features to build a regression model that predicts the concentration of toxic metals from the observed spectra. The limits of detections (LOD) were estimated using the detected features and compared to the traditional single-feature approach. LASSO is capable of detecting highly sensitive features in the input spectrum; however for some toxic metals the single-feature LOD marginally outperforms LASSO LOD. The combination of low cost instruments with advanced machine learning techniques for data analysis could pave the path forward for data driven solutions to costly measurements.

# 1 Introduction

The United States Environmental Protection Agency (US EPA) lists a number of metals in their list of Hazardous Air Pollutants (HAPs). These metals are known or suspected to cause cancer or other serious health effects (Buzea et al. (2007); Pope III et al. (2002)). Table 1 lists the metals in US EPA's HAPs list. Table 2 lists other metals that are not on US EPA's HAPs list but have been implicated in a range of

Table 1: List of hazardous metals targeted by US EPA

| US EPA Metal HAPS |
| --- |
| Antimony |
| Arsenic |
| Beryllium |
| Cadmium |
| Chromium |
| Cobalt |
| Lead |
| Manganese |
| Mercury |
| Nickel |
| Selenium |

Table 2: List of other toxic metals

| Toxic Metals |
| --- |
| Copper |
| Iron |
| Zinc |

adverse health effects so are of concern to the California Air Resources Board (CARB). It has been shown that presence of these metals are associated with various health concerns such as diabetes (Zanobetti et al. (2009)), cardiovascular disease (Brook et al. (2004)), and asthma (Gent et al. (2009)). Therefore, it is necessary to monitor and quantify their ambient concentration.

Various techniques over the years have been developed and used to measure metal particles. X-ray fluorescence (XRF) (Van Meel et al. (2007); Vincze et al. (2002)) and inductively coupled plasma mass spectrometry (ICP-MS) (Rovelli et al. (2018); Venecek et al. (2016)) have been used traditionally to quantify metals in atmospheric particles. XRF is excellent for measuring lighter elements and metals on filter substrates, but for field application it is expensive, has a high limit of detection (LOD) for heavier elements, and includes radiation risk. ICP-MS requires collection of aerosol on a substrate, such as a filter or impactor foil, extraction of the metals or elements from the substrate using harsh acidic chemicals, and then analyzing in the ICP-MS along with standards that help the instrument quantitate. Moreover, ICP-MS is most suitable for heavier elements and metals so has a high LOD for lighter toxic metals and is not available in field-deployed, real-time applications. Additionally, these instruments are expensive and hence are limited by cost and complexity as well.

Spark-induced breakdown spectroscopy (SIBS) and laser-induced breakdown spectroscopy (LIBS) have been employed in various applications from combustion (Do and Carter (2013); Kiefer et al. (2012); Kotza-gianni et al. (2016)), nanomaterials (Davari et al. (2017a); De Giacomo et al. (2011); Hu et al. (2017); Matsumoto et al. (2015a,b, 2016)), and environmental/bio-hazards (Diwakar et al. (2012); Diwakar and Kulkarni (2012); Zheng et al. (2018b)), forensics (Martin et al. (2007)), semiconductors and thin films (Ax-ente et al. (2014); Davari et al. (2017b, 2019); Hermann et al. (2019)), explosives (Gottfried et al. (2009)), pharmaceuticals (Mukherjee and Cheng (2008a,b); St-Onge et al. (2002)), and biomedical (Abbasi et al. (2018); Baudelet et al. (2006); Davari et al. (2018)). Particularly, Fisher et al. (2001) studied various toxic metals in aerosols by optimizing the spectrometer response with respect to gate delay. Hunter et al. (2000)

employed spark emission spectroscopy for continuous monitoring of metallic elements in aerosols. Yao et al. (2018) used spark emission spectroscopy to obtain the carbon content of fly ashes. Diwakar and Kulkarni (2012) employed spark emission spectroscopy coupled with a corona aerosol microconcentrator (CAM) to improve the particle collection efficiency and detection limits of toxic metals. Zheng et al. (2017) characterized the CAM performance with respect to different experimental parameters and obtained the optimized design parameters for their CAM system.

Recently, machine learning and deep learning techniques have been applied in different fields. These techniques in general learn patterns that can be used to distinguish different labels. Boucher et al. (2015) employed various linear and nonlinear machine learning techniques on LIBS spectra obtained from geological samples and concluded that a combination of models yields a lower total error of prediction. Chengxu et al. (2018) used convolutional neural networks to detect potassium in LIBS spectra and improve the linearity of their prediction model incorporating deep convolutional layers. Zheng et al. (2018a) employed spark emission spectroscopy on metals and used partial least squares regression to analyze their spectra set. They compared their multivariate models to univariate models and showed in their study these two groups have similar performance.

While LIBS and SIBS address issues regarding the field measurement and instrument complexity, they are still considered expensive. Current interest in low-cost sensors and their ability to characterize local air pollution concentrations motivated development of a low-cost system. We employed two complementary approaches: (1) decreasing the cost of the electronics associated with SIBS and (2) incorporating advanced data analysis techniques to improve quantification and limit of detection. In recent years, numerous studies have used artificial neural networks (Ferreira et al. (2008)), partial least squares regression and least absolute shrinkage and selection operator (LASSO) (Dyar et al. (2012)) on emission spectra to improve the quantification and limit of detection of spectroscopic systems. In this study, we have developed a low-cost spark emission spectroscopy system to quantify toxic metals. To reduce the overall cost, inexpensive replacements for necessary components, such as the spark generator and delay generator have been developed in the lab. To improve performance, advanced machine learning tools such as K-Means clustering and LASSO have been employed to improve the system performance. The resulting instrument was evaluated against four toxic metals listed by US EPA.

# 2 Instrument development:

## 2.1 Spark generation system:

Setting up a spark emission spectroscopy system requires expensive components. However, depending on the application some of the components can be replaced. Components such as spark generator and delay generator can cost up to $10K and $5K respectively. According to our application and needs, we developed these components for less than $600 and $50 respectively. One costly component that is required for developing a spark emission spectroscopy system is the spark generation system. Numerous papers have studied the fundamental principles of spark emission spectroscopy (Sacks and Walters (1970); Walters (1969, 1977)). The key idea is to discharge a capacitor as quickly as possible to increase the power dissipated in the spark gap. Fig. 1 illustrates the schematic of the spark generation system. The overall goal is to charge a capacitor at high voltage and once it has been charged sufficiently, discharge the capacitor through the spark gap. An Arduino board controls the timing between charging and discharging the capacitor. A boost convertor converts 24v DC to 5000v DC and is connected to a mechanical relay with two switching states controlled with the Arduino board. In the charge state, the mechanical relay provides the conduction path between the boost convertor and the capacitor. In this configuration, the capacitor reaches full charge in 5s. Once the capacitor is fully charged, the Arduino board sends a signal to turn off the boost convertor and sends another signal to the mechanical relay to flip to the discharge state. At the discharge state, the mechanical relay provides a conduction path between the capacitor and the spark gap. Shepherd et al. (2000) showed that the discharge process could be controlled by a resistor after the spark gap. For low resistor values, the spark current exhibited a periodic behavior as the capacitor discharges, which can be associated with an under damped discharging. On the other hand, increasing the resistor value damped the discharge process and dissipated a large portion of the capacitor energy through the resistor instead of the spark gap.We found that 10 $\Omega$ resistor maximizes the power dissipation in the spark gap, while minimizing oscillations. Fig. 2

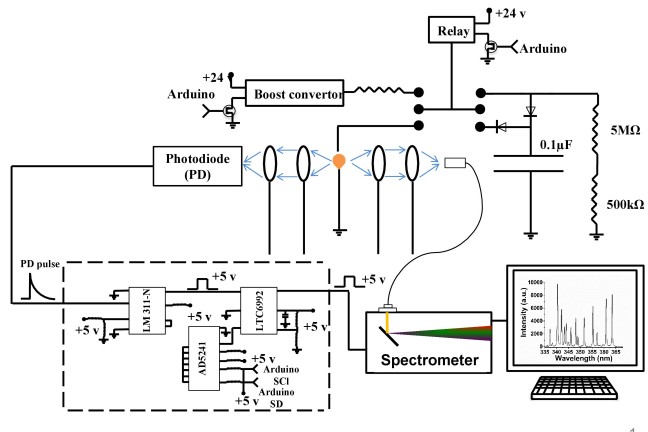

Figure 1: Schematic of the developed spark emission spectroscopy.

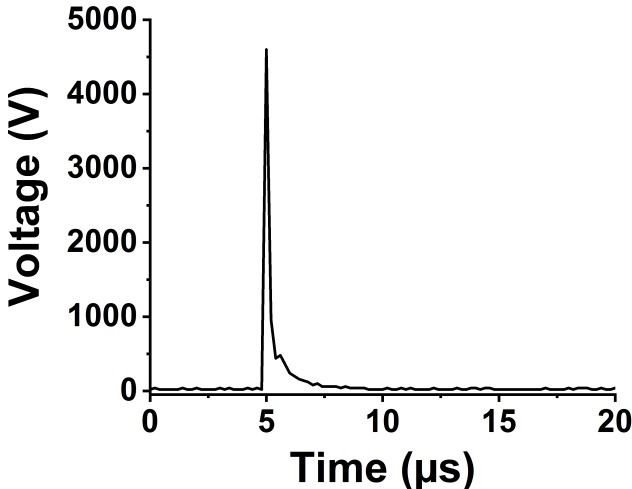

Figure 2: Spark voltage evolution in time.

illustrates the evolution of the generated spark as a function of time. The voltage shows a sudden increase followed by an exponential decrease fully discharging in less than $5\mu s$ and thus delivering sufficient energy to the arc and deposited analyte.

## 2.2 Delay generator:

The delay generator is another costly component typically used in time-resolved spectroscopy. Electronics advances have paved the way for developing a cost-effective delay generator. The delay generator suppresses initial noise in the emission spectrum so needs to cover a range between $1\mu s$ and $20\mu s$ with resolution less than $0.2\mu s$. We designed a custom-built delay generator in order to lower the overall cost of the instrument. Fig. 3 illustrates the schematic of the circuit. Upon generation of the spark-induced plasma, a pair of lenses collects and focuses the plasma emission into a photodiode. The pulse generated by the photodiode is passed into a voltage comparator (LM 311-N) to generate a transistor–transistor logic (TTL) signal. The output TTL signal from the comparator is sent to a pulse width modulator (PWM) controller (LTC6992), which adds delay to the TTL signal. An Arduino board adjusts a digital resistor (AD5241), which in turn determines the delay value. Fig. 4 shows the delay generator performance. The Y axis illustrates the delay values requested of the delay generator while the X axis shows the measured values. The red dashed line

shows the desired 1:1 line while the circles show the measured performance. The performance is linear over the relevant delay range with only a slight deviation from the 1:1 line. Considering the spark generated

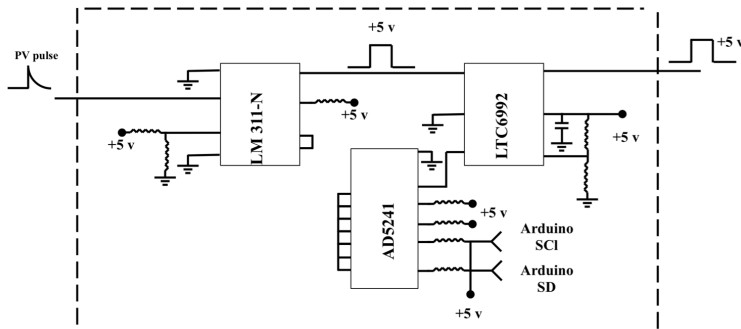

Figure 3: Schematic of the built-in delay generator.

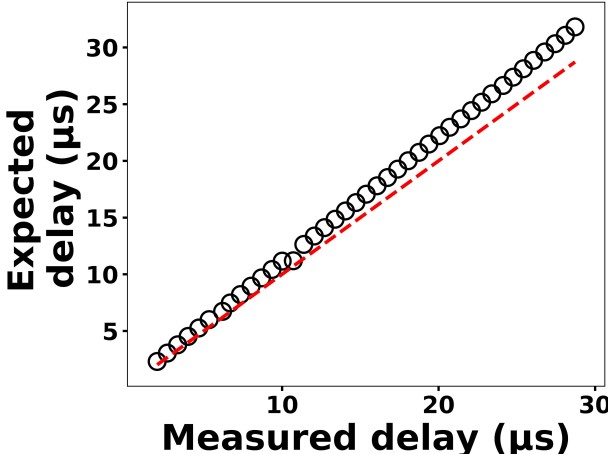

Figure 4: The expected delay set by the Arduino board as a function of the measured delay.

plasma short lifetime, our measurements require short delay values ($< 5\mu s$) where the built-in delay generator shows excellent performance and accuracy.

## 2.3 Spectra Collection:

Four toxic metals with different concentrations were used to test the developed spark emission spectrometer system performance. Cr, Cu, Ni and Pd ($1000\mu g$/mL) were purchased from AccuStandard and diluted to specific concentrations. For each concentration more than 10 spectra have been collected and used for model development. A micropipette was used to deposit diluted solutions on a 1 mm diameter Tungsten ground electrode of the spark system for emission analysis. The total mass can be calculated from the deposited volume and solution concentration. Upon evaporation of the droplets, the capacitor was discharged to ablate the deposited material and obtain spectra. A pair of lenses (75mm focal length and 1" diameter, Thorlab) focused the emission into an optical fiber connected to a spectrometer (Ocean Optics).

## 3 Results and discussions:

To address shot-to-shot variations in the spark-generated plasma and nullify possible faults caused by the low cost components, an unsupervised learning technique, K-Means clustering, classifies the collected spectra. Following this procedure, it is possible to identify and remove outliers and hence improve the accuracy of

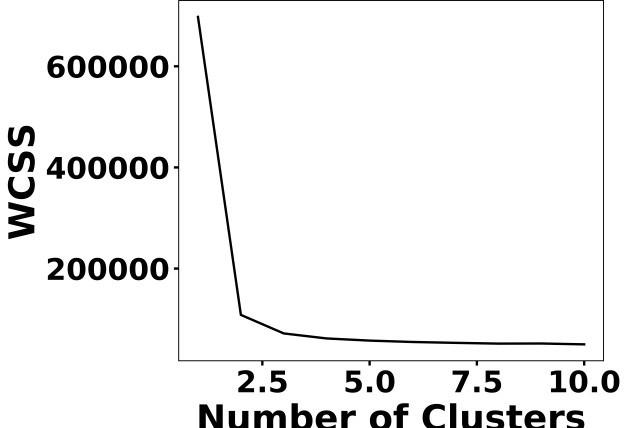

Figure 5: The elbow plot suggests two centroids for clustering the spectra set.

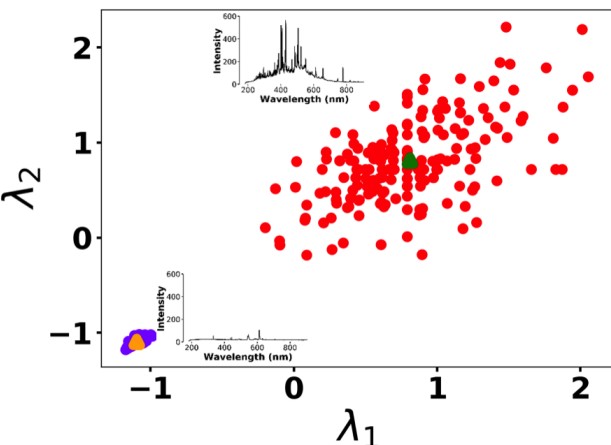

Figure 6: K-Means clustering for detecting outliers before passing the spectra set to LASSO model. Two clusters were plotted for the normalized intensities of two arbitrary wavelengths at $\lambda_1$ (208.365 nm) and $\lambda_2$g (208.759 nm).

the analysis. Fig. 5 illustrates the elbow plot that is used to optimize the number of spectral classes. The standard approach is to set the optimum number of clusters to the value where the within-cluster sum of squares (WCSS) error plateaus. The WCSS error plateaus once we have two or more centroids and therefore, the number of centroids is set to two. Fig. 6 illustrates the performance of the model for 300 spectra obtained from the background (Tungsten ground electrode ablation). The results show clearly two clusters with different emission response. The lower left cluster containing $< 10\%$ of the spectra represent low-signal outliers so were eliminated from further analysis. For each toxic metal, 0.1, 1, 10 and 100 ng of mass were deposited on the ground electrode. For each concentration, 10 spectra were collected using 2 $\mu$s delay between the observed and recorded emissions. After ablating the deposited mass and recording the spectrum, feature scaling has been used as a preprocessing step to improve the optimization process for out machine learning model. Plasma temperature can be obtained as:

$$I_{em} = \frac{hc}{\lambda_{ki}} N_k A_{ki} \tag{1}$$

$$N_k = N \frac{g_k e^{-\frac{E_k}{k_B T}}}{U(T)} \tag{2}$$

Combining equations (1) and (2) and taking log from both sides:

$$ln(\frac{I_{em}\lambda_{ki}}{g_k A_{ki}}) = -\frac{E_k}{k_B T} + ln(\frac{hcN}{U(T)}) \tag{3}$$

where $k_B$ is Boltzmann constant, $A_{ki}$ is the transition probability between two energy states (i) and (k), $N_k$ is the population density at energy state k ($E_k$). $\lambda_{ki}$ indicates the wavelength associated with the transition and $g_k$ represents the degeneracy of energy state k. The slope of equation (3) is used to estimate the plasma temperature based on a series of Tungsten lines for the recorded cleaned spectra set at $2\mu s$ . Fig. 7 illustrates the Boltzmann plot (Hahn and Omenetto (2010, 2012)) constructed by Tungsten lines. Based on the slope of the fit, the plasma temperature is estimated as $4013 \pm 579$ K. Upon identifying and removing the outlier

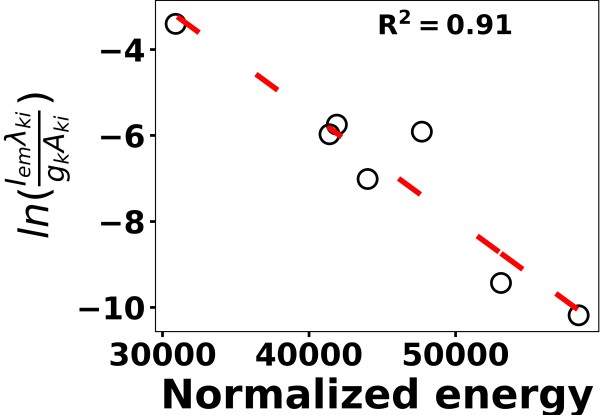

Figure 7: Boltzmann plot for various Tungsten lines in order to estimate plasma temperature.

spectra, the cleaned spectra set is normalized using the Tungsten peak at W I (400.87 nm) and fed into the Least Absolute Shrinkage and Selection Operator (LASSO) algorithm for model development and prediction.

## LASSO:

The cleaned scaled spectra set has been used to detect and quantify concentrations of the toxic metals. Simple linear regression obtains the slope and intercept of a linear line by minimizing the mean squared error between the predictions and known values. Least absolute shrinkage and selection operator (LASSO) detects and employs more features to perform predictions by optimizing the following loss function:

$$J(\boldsymbol{\theta}) = \frac{1}{m}\sum_{i=1}^{m}(y^{(i)} - h_\theta(\mathbf{x}^{(i)}))^2 + c\sum_{j=1}^{k}|\theta_j| \tag{4}$$

where $\mathbf{x}^{(i)} \in \mathbb{R}^{2048}$ and $h_\theta(\mathbf{x}^{(i)})$ represent the normalized spectrum and the LASSO concentration prediction based on spectrum (i) ($\mathbf{x}^{(i)}$), respectively, and where $y^{(i)}$ is the known concentration corresponding to spectrum (i). $m$ refers to total number of spectra and the LASSO coefficients are indicated by $\theta_j$. $k$ indicates the total number of features (spectral lines) used to build the model. The first term in equation (4) is the mean squared error and is common with simple linear regression, while the second term is a regularization term that minimizes the magnitude of $\theta_j$. The L1 norm essentially sets most of the features in the spectrum to zero and maintains only a few features to build the linear model and perform predictions. The regularization constant (c) determines the number of features to be used in the model, and therefore the model loss needs to be optimized with respect to the regularization constant. To obtain the optimized regularization constant, we plotted the loss values for the Ni spectra training and testing sets as a function of number of features for various c values based on Leave-One-Out cross validation (Fig. 8). As expected, the train loss monotonically decreases as the number of features increases, while the loss for the test set initially

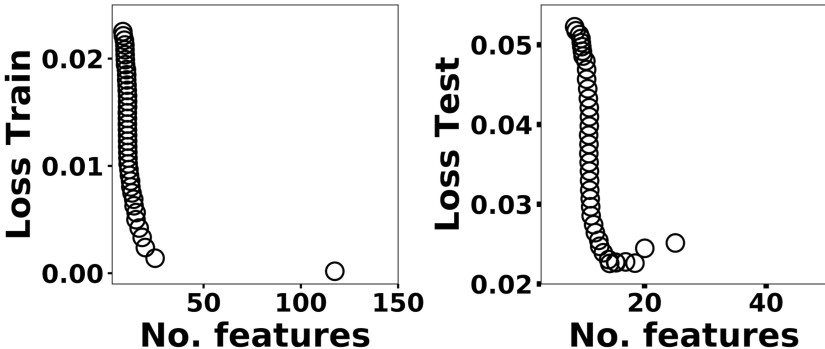

Figure 8: The train and test losses for Ni as a function of number of features.

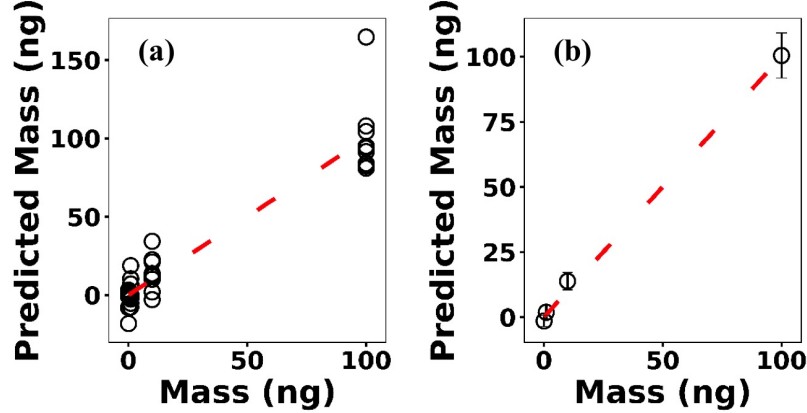

Figure 9: (a) LASSO predictions based on Leave-One-Out cross validation for Ni , (b) the averaged predictions for each concentration.

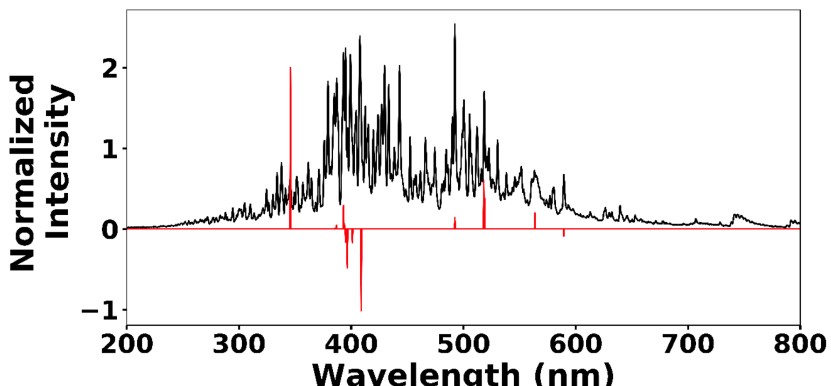

Figure 10: Ni 10ng spectrum (black line) and selected features by LASSO (red line).

decreases and then starts increasing. This implies that after incorporating a certain number of features into the model, the model starts memorizing rather than generalizing, which is known as overfitting. Therefore, we set the regularization constant to the value that minimizes the loss for the test set. Fig. 9 illustrates the optimized LASSO model predictions obtained by cross validation. For each concentration, the cross validation predictions were averaged and plotted along with the standard deviations. The predicted values vary linearly with the actuals. Figure 10 shows the wavelengths chosen by LASSO and the mean spectrum for 10 ng. LASSO chose a few Ni emission peaks along with other features to build the model. The same optimization process was applied to other toxic metals specifically Cr, Cu, and Pb. Fig. 11 illustrates the

resulting predictions and demonstrates the value of LASSO for predicting deposited mass from the spectra.

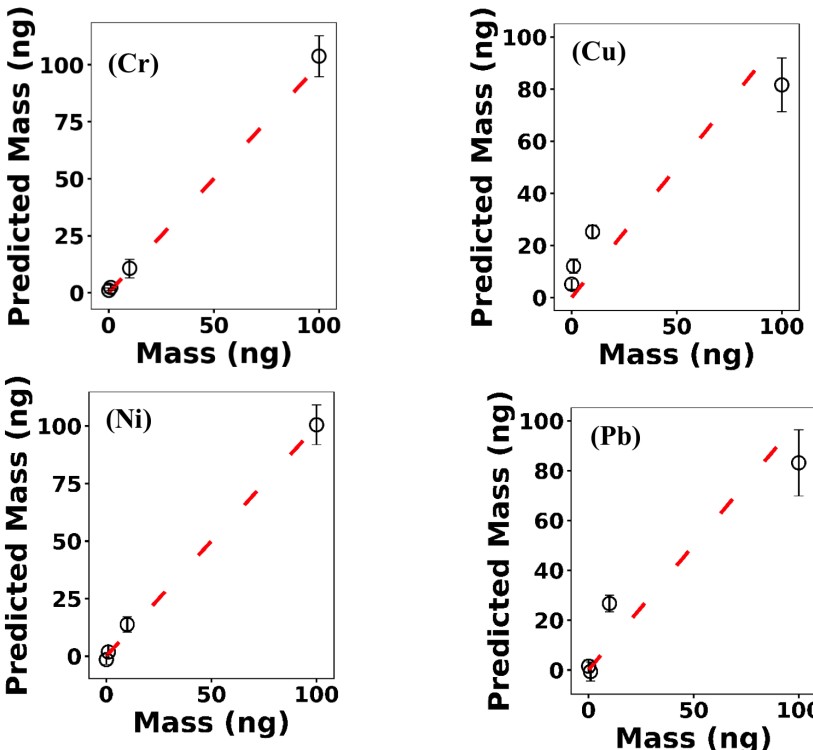

Figure 11: The optimized LASSO models predictions for Cr, Cu, Ni and Pb.

To obtain the limit of detection (LOD), the following function of the LASSO coefficients $\theta_j$ was used:

$$LOD = 3\frac{\sigma_B}{S} = 3\sigma_B \|\boldsymbol{\theta}_B\| \tag{5}$$

where $\sigma_B$ is the standard deviation of the background and $\|\boldsymbol{\theta}_B\|$ is the Euclidean norm of LASSO coefficients. Table 3 reports the LODs of the studied toxic metals.

Multivariate regression models such as LASSO might be more powerful in detection and quantification

Table 3: Detection limits for various toxic metals based on the LASSO and univariate models.

| Toxic Metal | LASSO | $R^2$ | MAE$_{\textbf{LASSO}}$ | Univariate | $R^2$ | MAE$_{\textbf{Univariate}}$ | Regularization cons. |
|---|---|---|---|---|---|---|---|
| Cr | 3.55 | 0.99 | 6.71 | 3.28 | 0.98 | 3.83 | 0.0008 |
| Cu | 12.09 | 0.92 | 49.67 | 0.68 | 0.11 | 143.27 | 0.0006 |
| Ni | 9.60 | 0.98 | 6.67 | 2.32 | 0.88 | 68.63 | 0.0009 |
| Pb | 54.40 | 0.90 | 36.67 | 8.37 | 0.45 | 124.42 | 0.0018 |

over univariate models; however, there is no guarantee that multivariate models outperform simple linear regression (Braga et al. (2010); Castro and Pereira-Filho (2016)). To evaluate LASSO performance, we compared LASSO with univariate methods, by calculating the LODs using simple univariate linear regression based on the features selected by LASSO. Fig. 12 illustrates the LODs obtained using this univariate technique (circles) compared to LASSO LOD (dashed line) for Ni. Considering only the sensitivity (LOD) is necessary but not sufficient for evaluating model performance since low $R^2$ values are also problematic. Therefore, in order to incorporate both $R^2$ and LOD for model assessment, we defined a score as:

$$Score = (\frac{LOD}{R^2})^2 \tag{6}$$

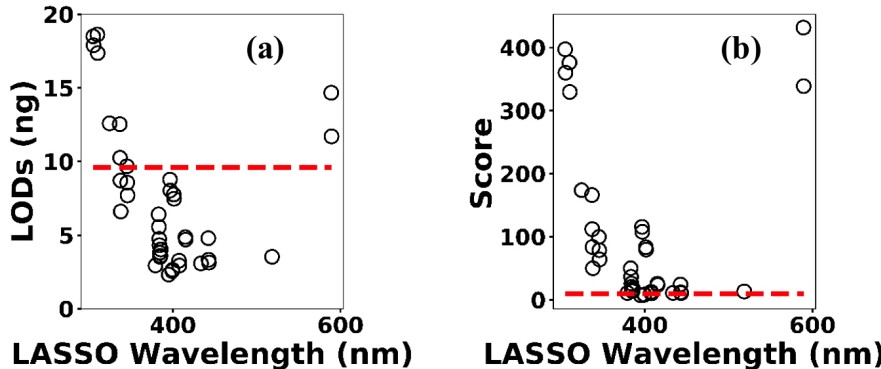

Figure 12: (a) the univariate LODs based on LASSO selected features and (b) LASSO and univariate models scores.

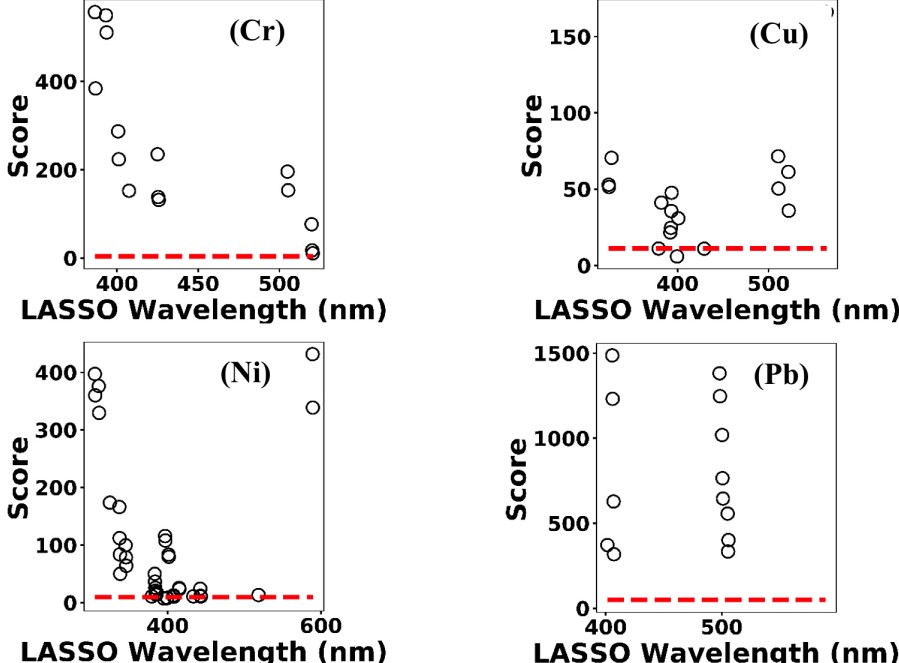

Figure 13: Model scores defined by equation 3 for Cr, Cu, Ni and Pb. Circles indicate univariate models scores and dashed lines correspond to LASSO scores.

Based on this definition, a model that has low LOD and high $R^2$ is desirable. LASSO score outperforms single feature linear regression for Pb, but the two methods were comparable for Cu, Ni, and Cr (Fig. 13). Other studies have reported that univariate techniques performed better than multivariate ones (Braga et al. (2010); Castro and Pereira-Filho (2016)). In LASSO, this may be related to the cost function defined for the regression (equation (4)). LASSO is a special case of elastic net family where both L1 and L2 norms are combined and used in the cost function. Considering the cost function in equation (4), the model goal is to minimize the prediction error and coefficient values (minimizing L1). This does not necessarily optimize LOD. Therefore, cost function minimization does not correspond to LOD minimization. Considering Fig. 12, using features defined by LASSO in a univariate model may yield better LOD than that obtained by LASSO alone. This might be an advantageous approach if the physical intuition of the features is not as important as detection of toxic metals.

# 4  Conclusion

A cost-effective spark emission spectroscopy instrument was designed and developed to quantify toxic metals targeted by US EPA and the California Air Resources Board. Costly components such as the spark generation system and delay generator were developed to lower the overall cost. An unsupervised learning technique was employed to detect outlier spectra. The cleaned spectra set was fed into LASSO for predicting the concentration of deposited samples on the ground electrode of the spark system from spectra obtained from the plasma. A combination of LASSO feature detection with univariate regression might improve the detection limits. Our results illustrate the promising realm of cost-effective sensors combined with advanced machine-learning techniques to provide data driven solutions to the traditional challenging problems.

# Funding

California Air Resources Board (CARB).

# Disclosures

The authors declare no conflicts of interest.

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
