# Peer review of "Quantification of toxic metallic elements using machine learning techniques and spark emission spectroscopy"

_Atmospheric Measurement Techniques, 2019_

## Referee Comment (RC1) · Anonymous Referee #1 · 6 Dec 2019

My major concern is that quantification of deposited samples on the electrode should be difficult. The accurate known amount of deposited mass will guarantee the improvement of quantitative performance of the spark system. How to control the mass of deposited elements on a 1 mm diameter tungsten ground electrode of the spark system for emission analysis ? The use of a micropipette was sufficient to control the mass of certain element on the electrode ?

Is this spark system applicable for the ambient PM samples, which should be their target in the future as stated in introduction ? The airborne particle deposition on the electrode would be totally different from current configuration where the solution was

deposited with the use of micropipette.

Most of contents in this paper were related to machine learning process for analysis of emission spectra, not much about aerosol measurement systems. In my view, it might be more appropriate to submit this study to machine learning or AI journals.

Minor things

More detailed schematics for experimental setup should be useful. Figures 1 and 3 would be moved to supplementary materials.

"For each element, 0.1, 1, 10 and 100 ng of mass were deposited on the ground electrode. For each concentration, 10 spectra were collected using 2 $\mu$g delay between the observed and recorded emissions." How to obtain the mass here ? More explanation is required here. The delay time unit would be us, not ug.

Any effect of delay time control on the emission spectra ?

More explanation on the spark energy? Can you guess plasma temperature? How stable the spark energy ? You may discuss this with laser induced plasma system

---

## Referee Comment (RC2) · Anonymous Referee #2 · 19 Dec 2019

A new approach is presented to detect toxic metallic elements in the atmosphere. The approach is based on spark emission spectroscopy. The authors develop a new spectrometer that they claim is more cost-effective than previous detectors. They record spectra for Cr, Cu, Ni and Pb at different concentrations and then deploy the Least Absolute Shrinkage and Selection Operator (LASSO) machine learning method. It is still not clear to me why they apply LASSO, presumably to calibrate their spectrometer.

The approach seems interesting and promising. However, the manuscript is quite unstructured and I had to deduce the main objective since it is not clearly stated. The manuscript is quite immature and in its current state not suitable for publication. There

is no flow and logical connection between sections and results are reported out of place and in the wrong order. Maybe some of the statements are clear to members of the atmospheric community, but they were not clear to me.

More detailed comments are given below.

- Table 1 lists hazardous elements with their full names, whereas Table 2 only gives the chemical elements. For consistency this should be changed. Also, chromium, nickel and lead show up in both tables, which leaves only copper as unique element in Table 2. This is strange.

- "Table 2 lists other metals that are not on US EPA's HAPs list but have been implicated in a range of adverse health effects so are of concern to the California Air Resources Board (CARB). X-ray fluorescence (XRF) and inductively coupled plasma mass spectrometry (ICP-MS) have been used traditionally to quantify metals in atmospheric particles." This is just one example of the structural and systemic problems of this article. From a list of hazardous elements the authors jump straight, in the same paragraph, to detection methods for such elements without establishing first that an important task or objective is to measure metal elements in the atmosphere. Maybe I am being picky here, but I had difficulties reading the introduction and following the logic.

- "LOD": the abbreviation LOD in the introduction is not explained

- The introduction essentially consists of an endless list of previous studies. At the end the reader is none the wiser, because no assessment or reflection is given. Worse; after this endless and tedious list the authors say "In this study we employed spark emission spectroscopy to quantify toxic metallic elements." At this point the readers wonders "so what?". Another methods for detection. What is new?

- The introduction does not mention LASSO or machine learning at all. Since this is quite a large part of the paper, it should be mentioned. There are plenty of good

overview articles for machine learning to refer to, for example.

- When asked to review the manuscript, my interest was piqued by the prospect of machine learning for spectra. Most machine learning is done for single target properties and machine learning for spectra is quite difficult. Examples are:

J. Timoshenko, D. Lu, Y. Lin, A. I. Frenkel, J. Phys. Chem. Lett. 2017, 8, 5091.

J. Timoshenko, A. Anspoks, A. Cintins, A. Kuzmin, J. Purans, A. I. Frenkel, Phys. Rev. Lett. 2018, 120, 225502.

C. Zheng, K. Mathew, C. Chen, Y. Chen, H. Tang, A. Dozier, J. J. Kas, F. D. Vila, J. J. Rehr, L. F. J. Piper, K. A. Persson, S. P. Ong, npj Comput. Mater. 2018, 4, 12.

K. Ghosh, A. Stuke, M. Todorović, P. B. Jørgensen, M. N. Schmidt, A. Vehtari and P. Rinke, Adv. Sci. 6, 1801367 (2019)

A. Cui et al Phys. Rev. Applied 12, 054049 (2019)

W. Lee, A. T. M. Lenferink, C. Otto, H. L. Offerhaus, J. Raman. Spectrosc. 1 (2019)

But then I saw no discussion of machine learning for spectra in this article.

- Instrument development: this part appeared strange and out of place to me on my first reading of the manuscript. At no point had I been prepared for a long, technical description of a new spectrometer. I think this is mostly a flow and logic problem again that can be solved by having a few connecting sentences that guide the reader through the paper.

- Figure 4 shows the expected delay as a function of the measured delay. I don't quite understand what that tells me or why that is important, but first of all, what are the circles in Figure 4 and what is the red dashed line. Second, how does one get from this expected or measured delay to a spectrum as shown in Figure 9?

- "an unsupervised learning technique, K-means clustering..." how is the K-means

applied? No details are given.

- It is not clear to me why a clustering technique is applied in the first place. The authors say this is to remove outliers. But what outliers? How do they manifest and why are they there in the first place? Should one remove them even?

- The results section then jumps straight to LASSO without saying why LASSO is applied. It is still not 100% clear to me. What is actually measured by the spark emission spectrometer and why does one need machine learning?

- Figure 6 has two insets that are way too small to be readable. Moreover, I do not understand what I see in the Figure and the caption is confusing. The points (how have they been determined) seem to cluster in a red region and a purple region that is barely visible. What does this mean and what is then done with that information?

- Equation 1 and its explanation make no sense to me. What is x? The discretised x-axis, in other words the wavelength values? h_theta is apparently the normalised spectrum, but why does it depend on the LASSO coefficients theta? y(i) is the known concentration corresponding to spectrum i, but in equation 1 h_theta is subtracted from y(i). How can a spectrum be subtracted from a concentration?

- "Therefore, we set the regularization constant to the value that minimizes the loss for the test set." And what is that value? It should be reported. C is a hyper parameter and hyper parameters are an essential part of machine learning.

- Figure 9 now suddenly shows a spectrum. More like an afterthought. How would one actually extract the mass from such a spectrum?

- Machine learning features heavily in the manuscript. However, at no point do the authors demonstrate that their method actually learns, i.e. its accuracy improves with more data. It is now standard to show learning curves in machine learning work. A learning curve plots the target (e.g. the prediction accuracy) as a function of training data size. The predictive accuracy should increase with increasing training set size (i.e.

the error in the prediction decreases).

---

## Author Comment (AC1) · 17 Feb 2020

1. My major concern is that quantification of deposited samples on the electrode should be difficult. The accurate known amount of deposited mass will guarantee the improvement of quantitative performance of the spark system. How to control the mass of deposited elements on a 1 mm diameter tungsten ground electrode of the spark system for emission analysis? The use of a micropipette was sufficient to control the mass of certain element on the electrode?

Authors' Response: First of all, the authors would like to thank the reviewers for their time and effort in evaluating the manuscript. As the reviewer accurately mentioned,

the amount of deposited material would change the spark-induced plasma response. In our experiments, we used a $10\mu$L pipet and added the solution with known concentration in small increments. This would assist us to control the deposition process in order to minimize solution loss to electrode's wall. The choice of $10\mu$L pipet was not obvious. Many experiments were performed with many pipette configurations before settling on this choice. Using large pipets make the deposition control difficult, and hence introduces errors in the amount of deposited mass.

2. Is this spark system applicable for the ambient PM samples, which should be their target in the future as stated in introduction? The airborne particle deposition on the electrode would be totally different from current configuration where the solution was deposited with the use of micropipette.

Authors' Response: The authors agree with reviewer. At this stage, our focus was to develop a low-cost core of an instrument that can be used to detect and quantify toxic metals in atmosphere. The current set-up can be employed to characterize ambient PM by adding a nozzle to deposit particles onto the electrode. Previous studies have shown the feasibility of such a design (Diwakar and Kulkarni 2012). Compared to those studies, we aimed to illustrate the possibility of setting-up similar systems at low-cost. We accomplished this goal by designing and prototyping the expensive components such as spark generation, delay generator, and controlling unit.

3. Most of contents in this paper were related to machine learning process for analysis of emission spectra, not much about aerosol measurement systems. In my view, it might be more appropriate to submit this study to machine learning or AI journals.

Authors' Response: The main reason that we incorporated two machine learning (ML) algorithms in our study was to address issues related to low-cost components and improve the system performance. Indeed, here our main objective is to introduce a low-cost system that can detect and quantify toxic metals in air in an affordable fashion. However, we want to illustrate the potentials that these ML techniques have for the

community. With the progresses in cloud-based platforms and AI, it is essential to adopt these technologies in fields where they can be beneficial. In the work presented here, we have not developed new ML algorithms. Instead, we have employed standard algorithms on the data generated by this system in order to predict concentrations.

More detailed schematics for experimental setup should be useful. Figures 1 and 3 would be moved to supplementary materials.

Authors' Response: The main reason that we provided those figures was to show the low-cost perspective of the paper and the possibility of designing a low-cost set-up for such a measurement.

"For each element, 0.1, 1, 10 and 100 ng of mass were deposited on the ground electrode. For each concentration, 10 spectra were collected using 2 $\mu$g delay between the observed and recorded emissions." How to obtain the mass here ? More explanation is required here. The delay time unit would be $\mu$s, not $\mu$g.

Authors' Response: The authors would like to thank the reviewer for correcting our error. This has been addressed in the revised manuscript at Pg. 6, Line 126: "For each element, 0.1, 1, 10 and 100 ng of mass were deposited on the ground electrode. For each concentration, 10 spectra were collected using 2 $\mu$s delay between the observed and recorded emissions." We purchased standard samples with known concentrations and then diluted them to known concentrations. By taking specific volume out of the diluted solution, we were able to calculate the mass that was deposited on the ground electrode. In order to address the reviewer comment, we have added the following at Pg. 5, Line 112: "The total mass can be calculated from the deposited volume and solution concentration."

Any effect of delay time control on the emission spectra?

Authors' Response: Previous studies including the authors' (Davari, Hu et al. 2017, Davari, Hu et al. 2017, Davari, Masjedi et al. 2017) have shown that by increasing the

delay time, the continuum emission decreases. At the same time the plasma species emissions also decreases, but for some species it is possible that the continuum decreases more quickly than the signal and hence improves the signal-to-noise ratios. In this study, we have chosen $2\mu$s for all measurements. Since the optimum delay varies based on the element of interest, it would not be practical to change the delay time especially if we do not know the type of element in advance. Our main purpose was to have a system with performance that is independent of the delay, independent of knowing the type of elements of interest in advance. Having a short delay will suppress the elements signal, and a long delay will lose information and intensity level. Moreover, it has been shown that usually after $1\mu$s the local thermodynamic equilibrium is established, which assists us to diagnose the plasma characteristics.

More explanation on the spark energy? Can you guess plasma temperature? How stable the spark energy ? You may discuss this with laser induced plasma system.

Authors' Response: From Lochte-Holtegreven (Lochte-Holtgreven 1968) and based on the Boltzmann distribution, the emission from an energy level k is equal to:

$$I_{em} = h\nu_{ki}N_kA_{ki} \tag{1}$$

$$N_k = N\frac{g_k e^{-\frac{E_k}{k_B T_{exc}}}}{U(T_{exc})} \tag{2}$$

Replacing Nk from equation (2) into (1) gives:

$$I_{em} = \frac{hcN}{U(T_{exc})}\frac{g_k A_{ki}}{\lambda_{ki}}e^{-\frac{E_k}{k_B T_{exc}}} \tag{3}$$

Therefore, taking the logarithm of both sides, we obtain:

$$ln(\frac{I_{em}\lambda_{ki}}{g_k A_{ki}}) = -\frac{E_k}{k_B T_{exc}} + ln(\frac{hcN}{U(T_{exc})}) \tag{4}$$

Considering various transition of a species, one can plot $ln(\frac{I_{em}\lambda_{ki}}{g_k A_{ki}})$ as a function of normalized upper energy level $\frac{E_k}{k_B}$. Based on equation (4), the slope of the linear fit to the Boltzmann plot (a) indicates $\frac{-1}{k_B T_{exc}}$. Therefore, the plasma excitation temperature is obtained as:

$$T_{exc} = -\frac{1}{a} \tag{5}$$

Following the procedure using the Tungsten lines:

---

## Author Comment (AC2) · 17 Feb 2020

A new approach is presented to detect toxic metallic elements in the atmosphere. The approach is based on spark emission spectroscopy. The authors develop a new spectrometer that they claim is more cost-effective than previous detectors. They record spectra for Cr, Cu, Ni and Pb at different concentrations and then deploy the Least Absolute Shrinkage and Selection Operator (LASSO) machine learning method. It is still not clear to me why they apply LASSO, presumably to calibrate their spectrometer. The approach seems interesting and promising. However, the manuscript is quite unstructured and I had to deduce the main objective since it is not clearly stated. The

manuscript is quite immature and in its current state not suitable for publication. There is no flow and logical connection between sections and results are reported out of place and in the wrong order. Maybe some of the statements are clear to members of the atmospheric community, but they were not clear to me.

Authors' Response: The authors would like to thank the reviewer for his/her time during the peer-review process. We also would like to state that we have not developed a detector. The paper presents our investigation for development of a spark emission spectroscopy system that uses Ocean Optics spectrometer as the system detector. Moreover, LASSO is a regression technique that has been used for data analysis, not detector calibration.

2. Table 1 lists hazardous elements with their full names, whereas Table 2 only gives the chemical elements. For consistency this should be changed. Also, chromium, nickel and lead show up in both tables, which leaves only copper as unique element in Table 2. This is strange.

Authors' Response: The authors thank the reviewer for pointing out the error. In the revised manuscript we have addressed the reviewer comment.

3. "Table 2 lists other metals that are not on US EPA's HAPs list but have been implicated in a range of adverse health effects so are of concern to the California Air Resources Board (CARB). X-ray fluorescence (XRF) and inductively coupled plasma mass spectrometry (ICP-MS) have been used traditionally to quantify metals in atmospheric particles." This is just one example of the structural and systemic problems of this article. From a list of hazardous elements the authors jump straight, in the same paragraph, to detection methods for such elements without establishing first that an important task or objective is to measure metal elements in the atmosphere. Maybe I am being picky here, but I had difficulties reading the introduction and following the logic.

Authors' Response: The authors would thank the reviewer for the directions to improve

the manuscript. In the revised manuscript, we have addressed the issue to improve the introduction readability.

-"LOD": the abbreviation LOD in the introduction is not explained.

Authors' Response: The authors have added the definition of LOD in Pg. 2, Line 37: ". . . is expensive, has a high limit of detection (LOD) for heavier elements,. . ."

The introduction essentially consists of an endless list of previous studies. At the end the reader is none the wiser, because no assessment or reflection is given. Worse; after this endless and tedious list the authors say "In this study we employed spark emission spectroscopy to quantify toxic metallic elements." At this point the readers wonders "so what?". Another methods for detection. What is new?

Authors' Response: The main goal of this study is to develop a low-cost spark emission system and improve the analytical performance using advanced data analysis techniques such as K-Means clustering and machine learning. In order to address the reviewer concern, we have added the following at Pg. 3, Line 59: 'While LIBS and SIBS address issues regarding the field measurement. . .'

The introduction does not mention LASSO or machine learning at all. Since this is quite a large part of the paper, it should be mentioned. There are plenty of good overview articles for machine learning to refer to, for example. - When asked to review the manuscript, my interest was piqued by the prospect of machine learning for spectra. Most machine learning is done for single target properties and machine learning for spectra is quite difficult. Examples are: J. Timoshenko, A. Anspoks, A. Cintins, A. Kuzmin, J. Purans, A. I. Frenkel, Phys. Rev. Lett. 2018, 120, 225502. C. Zheng, K. Mathew, C. Chen, Y. Chen, H. Tang, A. Dozier, J. J. Kas, F. D. Vila, J. J. Rehr, L. F. J. Piper, K. A. Persson, S. P. Ong, npj Comput. Mater. 2018, 4, 12. K. Ghosh, A. Stuke, M. Todorovic, P. B. Jorgensen, M. N. Schmidt, A. Vehtari and P. Rinke, Adv. Sci. 6, 1801367 (2019) A. Cui et al Phys. Rev. Applied 12, 054049 (2019) W. Lee, A. T. M. Lenferink, C. Otto, H. L. Offerhaus, J. Raman. Spectrosc. 1 (2019) But then I saw no

discussion of machine learning for spectra in this article.

Authors' Response: The authors would like state that the main focus of the current research was development of a low-cost spark emission spectroscopy to detect and quantify toxic metal PMs in atmosphere. Compared to the pervious studies, the expensive components such as spark generation and delay generator have been developed by the authors. The low-cost components such as delay generator might show false readings in some instances. We employed advanced machine learning techniques such as K-Means clustering to detect those false readings and discard them in order to clean the spectroscopic dataset and consequently reduce the errors. Moreover, most of metallic transitions occur in UV-VIS region of the spectrum. Using a low-cost spectrometer, we would not be able to resolve the spectrum sufficiently to detect individual metallic peaks to use them for quantification. Therefore, it becomes challenging to identify features in the spectrum that might be used for quantification. Instead, we chose LASSO as our data analysis technique. LASSO has the advantage that is not limited to individual peaks and performs the feature selection automatically and hence more suitable for identify metallic elements. In order to address the reviewer comment, we have added the following in Pg. 3, Line 59: "While LIBS and SIBS address issues regarding..."

Instrument development: this part appeared strange and out of place to me on my first reading of the manuscript. At no point had I been prepared for a long, technical description of a new spectrometer. I think this is mostly a flow and logic problem again that can be solved by having a few connecting sentences that guide the reader through the paper.

Instrument development: this part appeared strange and out of place to me on my first reading of the manuscript. At no point had I been prepared for a long, technical description of a new spectrometer. I think this is mostly a flow and logic problem again that can be solved by having a few connecting sentences that guide the reader through the paper.

Authors' Response: As it was mentioned, the main goal of this study is to develop a low-cost SIBS system and hence the authors provided the details related for instrument development. In order to address the flow of the manuscript, the authors have added the following in Pg. 3, Line 59: "While LIBS and SIBS address issues regarding..."

- Figure 4 shows the expected delay as a function of the measured delay. I don't quite understand what that tells me or why that is important, but first of all, what are the circles in Figure 4 and what is the red dashed line. Second, how does one get from this expected or measured delay to a spectrum as shown in Figure 9?

Authors' Response: In time resolved spectroscopy, usually a delay generator is needed to resolve the spectrum temporarily. We have designed and developed a delay generator to reduce the cost. Figure 4 shows the performance of our delay generator. The Y axis illustrates the delay values that we set with the delay generator, and X axis shows the delay values that we measured using an oscilloscope. The circles indicate the measured values with the oscilloscope and the red dash line indicates the one-to-one ratio line. To clarify the Figure, we have added the following at Pg. 4, Line 103: "Fig. 4 shows the delay generator performance. The Y axis illustrates the delay values requested of the delay generator while the X axis shows the measured values. The red dashed line shows the desired 1:1 line while the circles show the measured performance. The performance is linear over with a slight deviation from the 1:1 line."

"an unsupervised learning technique, K-means clustering. . ." how is the K-means applied? No details are given.

Authors' Response: The K-Means clustering performed in order to discard the "outliers" from the spectra dataset. As it was stated in the manuscript, the first step is to determine the number of clusters. This has been performed by plotting the within-cluster sum of squares (WCSS) as a function of number of clusters. Obviously, increasing the number of clusters will reduce the error. However, this might lead to overfitting problem. It is standard to set the optimum number of clusters to the value, where WCSS error

becomes plateau. This has been shown in Figure 5. In order to address the reviewer concern, the following has been added in the revised manuscript at Pg. 5, Line 120: "The standard approach is to set the optimum number of clusters to the value where the within-cluster sum of squares (WCSS) error plateaus."

"The standard approach is to set the optimum number of clusters to the value where the within-cluster sum of squares (WCSS) error plateaus."

Authors' Response: The spark emission spectroscopy is based on ablating materials using high voltage-current system. Since the voltage and current are very high, they create electromagnetic interference that might affect the delay generator and other electronic components. This results in noise in the electroncis and hence generates outliers in the dataset. To address this issue, we employed K-Means clustering to identify outliers in the dataset. For example, the following graph shows the spectrum of Cr obtained after $2\mu$s delay: However, if electromagnetic fields generated by the spark interferes with the electronics altering the delay value, the following spectrum results: as it can be observed the second spectrum is completely different from the normal spectrum and incorporating the second in our analysis will add error to the further analysis. K-Means clustering ensures us that the cleaned dataset will not contain erroneous spectra thus improving the accuracy and precision of the linear models.

The results section then jumps straight to LASSO without saying why LASSO is applied. It is still not 100% clear to me. What is actually measured by the spark emission spectrometer and why does one need machine learning?

Authors' Response: The goal of the study is to detect and quantify toxic metal concentration in atmosphere. The spark generates plasma that excites toxic metals. Once they relax back to ground states they emit the orbitals energy difference as light. A fiber optics collects the light and transmits it to a spectrometer that resolves the light into different wavelengths. We use the resulted spectrum to detect and quantify the concentration of toxic metal. In order to quantify the concentration of these pollutants,

we need to have a model that receives an input (i.e., a peak at a specific wavelength, multiple peaks, entire spectrum) and maps the input to the concentrations. This mapping can be generated using linear regression, neural networks, Gaussian process, etc. We chose LASSO as the model that receives the entire spectrum and maps it to concentration of the pollutants. LASSO compared to other techniques such as partial least square (PLS) can determine, which features (wavelengths) are more correlated to the output. This means that it only keeps a few features and discards the rest of features. In this study, our Ocean Optics spectrometer has 2048 pixels, which means that the recorded spectrum has 2048 features. Let's denote the entire spectrum as $\mathbf{x} \in^{2048}$. We can consider the entire spectrum as a high-dimensional vector. Our goal is to develop a mapping between this highly dimensional vector and concentration values:

$$h : \mathbf{x} \in^{2048} \to C \in \qquad (1)$$

LASSO compared to other regression models only uses a few features of the high-dimension vector to generate the linear model. It is worth mentioning that one of the main reasons to use ML was the spectrometer poor resolution. The current spectrometer does not have sufficient resolution to resolve close peaks. As a result, the peaks can convolute to each other and hence it is impossible to develop a model based on known emission peaks. In order to address the reviewer concern, the following has been added to the revised manuscript at Pg. 6, Line 139: "The cleaned scaled spectra set has been used to detect and quantify concentrations of the toxic metals."

Figure 6 has two insets that are way too small to be readable. Moreover, I do not understand what I see in the Figure and the caption is confusing. The points (how have they been determined) seem to cluster in a red region and a purple region that is barely visible. What does this mean and what is then done with that information?

Authors' Response: Figure 6 illustrates the effectiveness of K-Means clustering in detecting outlier spectra. As it was explained in the previous questions, K-Means clustering has been used to identify outlier spectra and exclude them from the LASSO. As

it was explained, each spectrum can be regarded as a high-dimensional vector, which each component of the vector indicates the intensity at a specific wavelength. The outcome of K-Means will be normal spectra set that has excluded the outliers from the spectra set.

Equation 1 and its explanation make no sense to me. What is x? The discretized x-axis, in other words the wavelength values? h theta is apparently the normalized spectrum, but why does it depend on the LASSO coefficients theta? y(i) is the known concentration corresponding to spectrum i, but in equation 1 h theta is subtracted from y(i). How can a spectrum be subtracted from a concentration?

Authors' Response: The followings summarize the terms:

- $x$ indicates the intensities correspond to each wavelength.

- $h : x \in^{2048} \to C \in$ The function with $\theta$ parameters (to be determined) that maps spectrum intensities to concentration.

- $y^{(i)}$ : The concentration corresponds to spectrum $i_{th}$.

To address the reviewer comment, the following has been added to the revised manuscript at Pg. 7, Line 140: '... $x^{(i)} \in^{2048}$ and $h_\theta(x^{(i)})$ represent...'

"Therefore, we set the regularization constant to the value that minimizes the loss for the test set." And what is that value? It should be reported. C is a hyper parameter and hyper parameters are an essential part of machine learning.

Authors' Response: The authors would like to thank the reviewer for pointing out the issue. In the revised manuscript the hyper parameters for various elements has been reported in Table 3.

Figure 9 now suddenly shows a spectrum. More like an afterthought. How would one actually extract the mass from such a spectrum?

Authors' Response: The mass is predicted based on the model that receives the spectrum as an input. Figure 10 illustrates the features that have been selected by LASSO model (red lines) and compares it with the original spectrum. The goal was to show how LASSO effectively chose less number of features and used them for developing a predictive model.

Machine learning features heavily in the manuscript. However, at no point do the authors demonstrate that their method actually learns, i.e. its accuracy improves with more data. It is now standard to show learning curves in machine learning work. A learning curve plots the target (e.g. the prediction accuracy) as a function of training data size. The predictive accuracy should increase with increasing training set size (i.e. the error in the prediction decreases)

Authors' Response: We actually have demonstrated the learning process in Figure 8. Figure 8 shows the loss value as a function of number of features. It is well known that as the number of features increases, the model over fit the data. The loss values for the training set indicate this phenomenon perfectly. Moreover, considering the loss values for the test set, we realize that the error increases after incorporating certain number of features, which suggests the optimum number of features.

---

## Author Response (AR2)

**Reviewer #1:**

Major concern is that this method would be difficult to be applied for detection of metallic components in atmospheric aerosols having low concentrations. Although their new machine leaning technique and development of low cost delay generator can be so promising, the feasibility of this technique to detect and/or to quantify atmospheric metallic components in PM should be still low, based on their current dataset (solution-based sample results using highly concentrated ones with uncertain amount of deposited or ablated mass).

Their deposition mass calculation is still unclear to readers.

1. Can authors be sure if their calculated mass is the same as the deposited mass on the 1 mm diameter tungsten ground electrode? The detailed experimental procedure with schematic should be required.

**Authors' Response:** The authors have used a 5µL pipette in order to carefully control deposition of sample on the electrode. Using a larger diameter pipette would have caused some the deposited mass stick to the Tungsten electrode circumference, reducing the mass on the electrode surface. Therefore, using this small pipette and slowly depositing mass on the electrode, assures that the calculated mass is similar to the deposited mass.

2. Their deposit samples on the 1 mm diameter tungsten ground electrode were ablated completely by the spark plasma (i.e., is it the same as the ablated mass)?

**Authors' Response:** What we present is a calibration-based technique in that we develop a calibration between the known mass deposited and the spectra obtained. There is no feasible way to measure the ablated mass, but this is accounted for in the calibration. We agree with the reviewer that the ablation process might change with the total deposited mass. However, there is substantial literature showing that such matrix effects should not be significant for the low amounts of mass that we are depositing and that will be deposited in the final instrument design.

3. I wonder if their calculated masses are relevant to the concentration range of metallic components in PM2.5 or PM10 in atmosphere.

**Authors' Response:** The concentration of the ambient toxic metals depends on where the instrument is tested. For industrial application, the concentration of metal PMs can reach to few $\mu g/m^3$. Much of the potential application of this instrument is in communities in proximity to sources that are potentially emitting a substantial amount of toxic metals. For the proposed instrument, the detection limit and sampling time are coupled together. Increasing the sampling time will increase the total collected mass on the ground electrode and hence improve the detection process.

**Reviewer #2:**

The authors have to some degree addressed my comments and the manuscript has improved. However, I still find it very hard to read, mostly because it has no clear structure and logical flow to it. Sections that describe developments, experiments, results and analysis are ultra short and leave out much detail. Much is left to the reader's imagination and interpretation, which is not a good scientific reporting style. The objectives of the work are still not entirely clear to me. Throughout the manuscript, the same phrases are repeated, but details on how these objectives are achieved is missing in the right places. Also, in my report on the original manuscript I have pointed out several of these shortcomings. The reply of the authors was to repeat the same phrases, which is not very impressive.

In conclusion, I am disappointed by the manuscript and by the authors' efforts to improve it. As somebody not immediately working in the field of atmospheric science, I accepted the review of the manuscript, because the title and the premise sounded interesting. I believe the manuscript is publishable, but it takes more than the current revisions to make the manuscript readable and understandable. More details are given below:

**Authors' Response:** The author would like to thank the reviewer for his/her valuable comments and we apologize if we did not address the prior comments sufficiently well. As mentioned before both in the manuscript and rebuttal, the objective of the current study is to develop a cost-effective instrument to detect toxic metallic elements in air. There are other techniques such as XRF and ICP-MS for such a task, however they both are expensive and difficult to handle in field measurements. The authors attempted to develop an instrument that addresses these shortcomings using low-cost components and improve the performance of the new instrument by employing advanced machine learning techniques.

- The introduction has now improved and it is easier to read. However, the section on the spark-induced breakdown spectroscopy (SIBS) and laser-induced breakdown spectroscopy (LIBS) is still very difficult to read. This is in part, because of the citation style. The citations are not separated from the actual text in any way and sometimes it is impossible to tell where a citation ends and the text starts again.

**Authors' Response:** We apologize and have changed the citation style to match that of the journal. We hope during the final production these citation style become more readable for readers.

- I had recommended to add more context and background to the introduction. The authors talk at length about different detection methods, but do not say much about machine learning in the introduction, although half of the manuscript is about machine learning. I had recommended to cite and then discuss several recent machine learning approaches for spectra and spectroscopy. Since then there have been even more in the literature. This request has been blatantly ignored by the authors. Makes me wonder why we reviewers write referee reports in the first place.

**Authors' Response:** The authors greatly appreciate the reviewer time and effort for providing useful inputs. In order to address the reviewer concern, the following has been added to the introduction starting at Pg. 3, Line 59:

"Recently…"

- The objectives are now stated more clearly: "We employed two complementary approaches: (1) decreasing the cost of the electronics associated with SIBS and (2) incorporating advanced data analysis techniques to improve quantification and limit of detection." However, what is still not clear to me is how the cost is decreased (see below). An example of frequent circular arguments in this manuscript is a sentence in the same paragraph in the introduction: "The expensive components such as spark generation and delay generator have been developed to reduce the overall cost." To me this sentence sounds like `we developed expensive components to reduce the cost', which would, of course, be a contradiction. I suppose the authors meant to say that they improved the expensive components so that they become cheaper. Maybe the sentence could also be improved.

**Authors' Response:** The following has been revised to address the reviewer comment at, Pg. 3, Line 75:

"To reduce the overall cost, inexpensive replacements for necessary components, such as the spark generator and delay generator have been developed in the lab"

- Both for the spark generation system and the delay generator, the manuscript states that these are expensive components. But it does not say why. What is expensive about them? I am not an expert on these instruments and most likely most readers will not be either.

**Authors' Response:** The cost of the spark generation system according the quote we received was ~$12K and delay generator ~$5K. At the beginning of this study, the authors asked the very same question and in fact that was the main motivation to develop these components in the lab. The cost of our spark system and delay generator are ~$600 and ~$50 respectively. The quoted spark system ($12K) can generate high-energy spark with high repetition rate (1Hz). However, considering low concentration of PM and the required time to sample enough materials makes this repetition rate completely unnecessary. For the commercial delay generators, they can generate delays from ps to ms range with high accuracy (fs). However, our measurements only require a few µs delay with accuracy of less than ~0.2µs. Therefore, purchasing a commercial delay generator also is unnecessary. Therefore, the authors decided to develop those components in the lab to reduce the overall cost. To address the reviewer comment, the following has been added at Pg. 3, Line 83:

"Setting up a spark emission spectroscopy system requires expensive components. However, depending on the application some of these components can be replaced. Components such as spark generator and delay generator can cost up to $10K and $5K respectively. Here according to our application and needs we developed these components for less than $600 and $50 respectively."

- How is the spark generator improved? The manuscript devotes literally only 1 sentence to this "In our setup, a 10Ω resistor maximizing power dissipation in the spark gap, while minimizing oscillations." And this sentence is not even grammatically correct. It has no verb. Such a short description is simply not good enough and not scientific at all.

**Authors' Response:** Having 0Ω resistor will cause oscillation in current and voltage in the spark gap. Therefore, it is necessary to add a resistance to damp these oscillations. However, adding too much resistance will increase the energy dissipated in the resistor and hence reduce the overall energy dissipated in the spark gap. The following has been revised at Pg. 4, Line 98:

"We found that 10 Ω resistor maximizes the power dissipation in the spark gap, while minimizing oscillations."

- Spectra collection: this section is also very short. How many spectra were collected in the end and at what settings? What is the spectral resolution? How are the spectra discretised, etc.? In other words, what is the data set?

**Authors' Response:** For each concentration more than 10 spectra have been collected and used for model development. Considering 4 sets of concentrations for each element and 4 elements in total, we collected 160 spectra for the entire study for each element. In order to address the reviewer question, the following has been added to the manuscript at Pg. 5, Line 123:

"For each concentration more than 10 spectra have been collected and used for model development."

- What I am missing throughout is a section and a schematic figure that illustrates the whole process. What does the instrument look like and how do the different components that the authors improve fit in? How do we go from spark generation to a spectrum? How is the spectrum further processed with machine learning? This schematic would ideally also illustrate the overall objectives of the research.

**Authors' Response:** In order to address the reviewer question, the asked schematic has been added to the revised manuscript at Pg. 3, Line 84.

- The results section has more figures than it has text. Very hard to understand anything.

**Authors' Response:** We have added some text in response to other reviewer comments. The figures are necessary to show what we found. We hope that the result is more acceptable to the reviewers.

- In the clustering section on page 6, I could not follow anything anymore after "For each element, 0.1, 1, 10 and 100 ng…" What does "element" refer to here? I think this part might now be describing data collection? Maybe? And therefore answer one of my earlier questions? But then seamlessly without paragraph break the paragraph continues with "Feature scaling is a standard preprocessing step…" How do we get from clustering to spectra collection to feature scaling? I cannot follow the logic anymore.

**Authors' Response:** The objective of this study was to detect and quantify toxic metallic PM in air. These PMs are comprised of different metallic elements as listed in the Table 1 and 2. "Element" is referring to the metallic elements such as Cu, Ni, etc that we used in this study to test our spark emission spectroscopy setup. After ablating the deposited mass and recording the spectrum, feature scaling has been used as a preprocessing step to improve the optimization process for out machine learning model. To address the reviewer concern, we have added the following at Pg. 6, Line 141:

"After ablating the deposited mass and recording the spectrum, feature scaling has been used as a preprocessing step to improve the optimization process for out machine learning model."

- "Figure 7 illustrates the Boltzmann plot…" What is a Boltzmann plot?

**Authors' Response:** Boltzmann plot is a well-known approach in plasma spectroscopy for obtaining plasma temperature. The authors explained how it is constructed in the previous rebuttal. Basically, the slope of the fit to the Boltzmann plot provides the plasma temperature. To address the reviewer comment the following has been added at Pg. 6, Line 146:

"Plasma temperature can be obtained as: …"

- Throughout the whole results section, the authors speak of features and feature selection. A "feature" in machine learning is an abstract concept that refers too patterns in the data input. It is ok to use the word features, but then it has to be clear what the data is, what the data input is to the machine learning and what the features are. I have figured out now, that the input into the machine learning is x-y-z data in the form of discretised spectra. x are the frequencies and y the intensities of the spectra and z are the detected masses of metals. A feature is then simply one point in the spectrum. This information may seem trivial to the authors, but without it, the manuscript is much harder to read and understand.

**Authors' Response:** The authors are glad that the manuscript can explain the model now. This information has been provided in Pg. 7, Line 155:

"where $x(i) \in R^{2048}$ and $h_\theta(x(i))$ represent the normalized spectrum and the LASSO concentration prediction based on spectrum (i) $(x(i))$, respectively, and where $y(i)$ is the known concentration corresponding to spectrum (i)."

- in equation 1 "m" appears to be the number of spectra, i.e. the data set size. Would be good to say that in the text.

**Authors' Response:** The following has been added at Pg. 7, Line 156:

"*m* refers to total number of spectra and …"

- Figure 8 is used to justify the choice of the regularisation constant c. I find this procedure also hard to follow. Instead of plotting the training and test loss as function of the number

of features (which is related to c) it would be more illustrative to plot it as a function of c. Or have an alternative x-axis at the top of the graph that shows the c values.

**Authors' Response:** The c constant does not provide much physical information while the number of feature provides the total number of spectrum features (lines) that has been used in the model. Here, feature refers to spectral lines in a spectrum. Hence, the total No. of feature indicates the total No. of spectral lines used to build the models. In order to clarify this, the following has been added at Pg. 7, Line 164:

"k indicates the total number of features (spectral lines) used to build the model."

- The whole discussion on the univariate "technique" is very hard to follow. What was actually done? What is plotted in Figures 12 and 13?

**Authors' Response:** Considering the vast interest toward machine learning techniques these days, the authors wanted to point out that machine learning is not the solution to everything and there is no guarantee that multivariate model always perform better than univariate model specially if the multivariate model has not trained efficiently. In order to address the reviewer comment, the following has been added at Pg. 9, Line 188:

"To evaluate LASSO performance, we compared LASSO with …"